# MPrompt: Exploring Multi-level Prompt Tuning for Machine Reading Comprehension

**Guoxin Chen**[★♦▲], **Yiming Qian**[♣], **Bowen Wang**[♠], **Liangzhi Li**[★▲*]

[★]Meetyou AI Lab    [♦]University of Chinese Academy of Sciences

[♣]Agency for Science, Technology and Research (A*STAR)    [♠]Osaka University

[▲]Xiamen Key Laboratory of Women's Internet Health Management

gx.chen.chn@gmail.com    qiany@ihpc.a-star.edu.sg

bowen.wang@is.ids.osaka-u.ac.jp    liliangzhi@xiaoyouzi.com

## Abstract

The large language models have achieved superior performance on various natural language tasks. One major drawback of such approaches is they are resource-intensive in fine-tuning new datasets. Soft-prompt tuning presents a resource-efficient solution to fine-tune the pre-trained language models (PLMs) while keeping their weight frozen. Existing soft prompt methods mainly focus on designing the input-independent prompts that steer the model to fit the domain of the new dataset. Those methods often ignore the fine-grained information about the task and context of the text. In this paper, we propose a multi-level prompt tuning (MPrompt) method for machine reading comprehension. It utilizes prompts at task-specific, domain-specific, and context-specific levels to enhance the comprehension of input semantics at different granularities. We also propose an independence constraint to steer each domain-specific prompt to focus on information within its domain to avoid redundancy. Moreover, we present a prompt generator that incorporates context-related knowledge in the prompt generation to enhance contextual relevancy. We conducted extensive experiments on 12 benchmarks of various QA formats and achieved an average improvement of 1.94% over the state-of-the-art methods[1].

## 1 Introduction

In recent years, pre-trained language models (PLMs) have been widely applied in question-answering tasks (Pandya and Bhatt, 2021), particularly in machine reading comprehension (Baradaran et al., 2022), and achieved remarkable success through the *pretrain-then-finetune* paradigm (Roberts et al., 2020; Khashabi et al., 2020b). Despite the excellent performance, due to the explosive growth of parameter sizes in PLMs,

the fine-tuning paradigm has become resource intensive.

Recently, soft-prompt tuning has been widely explored as a parameter-efficient approach to addressing the aforementioned issues (Liu et al., 2023). For example, Li and Liang (2021) proposed Prefix-tuning, which prepends a sequence of optimizable prefixes to each transformer layer while keeping the parameters of PLMs frozen. Prefix-tuning provides a lightweight alternative to fine-tuning and has achieved comparable performance with fewer trainable parameters. Lester et al. (2021) proposed Prompt-tuning, which only prepends optimizable prompt vectors to the input sequence, which used fewer parameters compared to Prefix-tuning. Ma et al. (2022) discovered negative tokens in Prompt-tuning that have a detrimental effect on downstream tasks and proposed XPrompt to mask these negative tokens, resulting in improved performance. However, the aforementioned methods are input-independent, i.e., assigning a uniform prompt to all inputs of a given task, which under-utilizes the input semantics for the answer generation in machine reading comprehension.

There is a growing trend towards designing input-dependent prompts (a.k.a dynamic prompts) for various tasks (Gu et al., 2021; Clive et al., 2022; Tang et al., 2022). For example, Gu et al. (2021) proposed DialogPrompt for a dialog system, which dynamically generates prompt vectors according to the input dialogue context. Tang et al. (2022) extracts input-related information from BERT (Devlin et al., 2018) as contextualized prompts for natural language generation (Lewis et al., 2019; Raffel et al., 2020), which improves the relevance between the generated text and the input text. However, to the best of our knowledge, there has been little research exploring input-dependent prompt methods for question-answering tasks, especially for machine reading comprehension. It is challenging to apply input-independent methods to ma-

---

[*]Corresponding author.

[1]The code is available at https://github.com/Chen-GX/MPrompt.

chine reading comprehension where the answer is context-sensitive.

To address the above issues, we propose **MPrompt**, a novel **M**ulti-level **Prompt** tuning approach for machine reading comprehension. Our method utilizes the dataset and the context information to create three levels of prompts: task-specific, domain-specific, and context-specific. The task-specific prompts are input-independent and generate a prompt based on the tasks. The domain-specific prompts utilize the domain knowledge generated from the dataset while context-specific prompts rely on the input context. These multi-level prompts endow PLMs with multiple fine-grained considerations of input semantics. To further enhance the domain-specific prompts and avoid information redundancy, we propose the independence constraint to steer each prompt to focus on knowledge within the domain rather than cross-domain knowledge. Furthermore, we extract context-related knowledge from a small-scale PLM, such as T5-small (Raffel et al., 2020), and integrate it into the prompt generation process to enrich the context sensitivity of prompts. With the help of these three levels of prompts, we achieve an average improvement of 1.94% over the state-of-the-art methods on 12 benchmark datasets.

Our main contributions are as follows:

- We propose a novel multi-level prompt tuning (MPrompt) for machine reading comprehension which generates prompts at task-specific, domain-specific, and context-specific levels to improve answer generation.

- We propose an independence constraint to steer each domain-specific prompt to focus on intra-domain information, avoiding information redundancy, at the same time enriching the domain-related semantics.

- We propose a prompt generator based on a small-scale PLM to integrate context-related knowledge into prompt generation, which enriches the context awareness and sensitivity of the generated prompts.

## 2 Related Work

### 2.1 Machine Reading Comprehension

Machine Reading Comprehension (MRC) is a challenging task and hot topic in Question Answering (QA) (Pandya and Bhatt, 2021; Baradaran et al.,

2022). It aims to comprehend contexts and provides answers to corresponding questions. In recent years, the focus of Machine Reading Comprehension research has shifted from Extractive Question Answering (Seo et al., 2016; Wang et al., 2017; Tan et al., 2018) to Generative Question Answering (Izacard and Grave, 2020; Khashabi et al., 2020b, 2022; Jiang et al., 2022). For example, Lewis et al. (2020) has explored a retrieval-augmented generation scheme that combined pre-trained retrieval models to enhance the performance of the generative question answering models. Khashabi et al. (2020b, 2022) unified the input format of different QA tasks into the same format and fine-tune the generative models (Raffel et al., 2020) for question answering. However, with the explosive growth in the parameter size of PLMs, the fine-tuning process becomes exponentially more resource intensive. One way to relax this computational requirement is through prompt learning (Li and Liang, 2021; Liu et al., 2023).

### 2.2 Prompt Learning

With the success of GPT-3 (Brown et al., 2020), prompt learning (Liu et al., 2023) has provided another efficient way to utilize PLMs, which has attracted widespread attention. The format of prompts can be in human-readable natural language (discrete prompts) (Shin et al., 2020; Schick and Schütze, 2020), or embedding vectors (continuous prompts) (Lester et al., 2021; Li and Liang, 2021; Liu et al., 2021a,b; Ma et al., 2022). The continuous prompts provide a more flexible solution that encodes information into a trainable embedding which presents the information to a pre-trained model more efficiently. For example, Lester et al. (2021) proposed Prompt-tuning, which achieves competitive performance by prepending trainable prompts to input sequences, and Ma et al. (2022) further improved the Prompt-tuning by pruning the negative prompt tokens.

The aforementioned approaches did not sufficiently consider the full utilization of the input semantics and applied the same prompt for all examples in the dataset, which potentially limits the delivery of the language models. Therefore, Tang et al. (2022) extracts contextualized prompts based on the input text from external PLMs, resulting in better performance in natural language generation. Clive et al. (2022) proposes to combine task-specific prompts with dynamic prompts, enabling

the model to have finer-grained control over the generated text.

However, there has been little research exploring input-dependent prompt learning in question answering. In contrast to natural language generation, question-answering tasks emphasize understanding of the given question and context. Therefore, a lack of input-dependent prompts may lead to an under-leverage of the context information present in addition to the questions, particularly in machine reading comprehension tasks.

## 3 Methodology

Our proposed multi-level prompt tuning (MPrompt) framework is illustrated in Figure 1. The framework consists of a prompt generator and a generative question answering model, whereas the former relies on a smaller-sized encoder-decoder architecture. The prompt generator generates domain-specific and context-specific prompts and elicits context-related knowledge from small-scale PLMs into the generation process.

### 3.1 Task-specific Prompt

Many previous works (Li and Liang, 2021; Lester et al., 2021) have demonstrated that shareable prompt parameters learned from particular tasks can effectively enhance the performance of pre-trained language models on downstream tasks. Therefore, following Li and Liang (2021), we construct task-specific prompts that share common prompt information within the task.

We prepend a prefix $P \in \mathbb{R}^{t \times d}$ for the different types of attention class in the pre-trained language models, where $t$ is the length of the task-specific prompt and $d$ is the dimension of the embedding in generative QA model. For each attention class[2], the prefix for key-value pairs $\mathcal{T} = \{\mathcal{T}_1, \mathcal{T}_2, ..., \mathcal{T}_L\}$ are learned through an MLP, $\mathcal{T} = \text{MLP}(P)$, where $L$ denotes the number of layers in the generative QA model, $\mathcal{T}_l = (\mathcal{T}_{l,K}, \mathcal{T}_{l,V}) \quad \forall l \in \{1, ..., L\}$, $\mathcal{T}_{l,K}$ and $\mathcal{T}_{l,V} \in \mathbb{R}^{t \times d}$, and $\mathcal{T} \in \mathbb{R}^{t \times 2dL}$. The overall task-specific prompt is $\mathcal{T}_{task} = \{\mathcal{T}_E, \mathcal{T}_{Dm}, \mathcal{T}_{Dc}\}$.

### 3.2 Domain-specific Prompt

In question answering scenarios, especially in machine reading comprehension, the context plays a

---

crucial role as it contains the answer or the evidence in support of the answer. Meanwhile, the context in QA datasets can often be divided into several domains. For example, in NewsQA (Trischler et al., 2016), the context can be grouped into different domains such as politics, economics, society, and so on. To improve the semantic understanding of context, the context from different domains should utilize different prompts, and each domain-specific prompt should imply a specific knowledge shared within the domain.

[2]In encoder-decoder architecture models, there are typically three types of attention: self-attention in the encoder, masked self-attention in the decoder, and cross-attention in the decoder. The corresponding task-specific prompts are denoted as $\mathcal{T}_E$, $\mathcal{T}_{Dm}$, and $\mathcal{T}_{Dc}$.

However, most QA datasets do not have explicit information about the domain of the context. To avoid additional annotation costs, we cluster the context C in an unsupervised manner to obtain different domains $D \in \{D_1, ..., D_n\}$, where $n$ denotes the number of domains, and each context can only belong to one domain. Each domain has its own shared prompt, therefore the domain-specific prompts $\mathcal{D} = \{\mathcal{D}_1, ..., \mathcal{D}_n\}$, where $\mathcal{D}_i \in \mathbb{R}^{\rho \times d_p} \quad \forall i \in \{1, ..., n\}$, $\mathcal{D}_i$ denotes the prompt shared within the domain $D_i$, $\rho$ denotes the length of the domain-specific prompts, $d_p$ denotes the dimension of embedding from the prompt generator.

Intuitively, domain-specific prompts should encapsulate information for each respective domain. Therefore, we introduce the independence constraint to steer $\mathcal{D}_i$ to focus on the information within domain $D_i$. Focusing on the knowledge specific to each domain can enhance contextual understanding, as confirmed by subsequent experiments. Specifically, for any pair of $\mathcal{D}_a$ and $\mathcal{D}_b \in \mathcal{D}$, we introduce the Hilbert-Schmidt Independence Criterion (HSIC) (Gretton et al., 2005; Song et al., 2007) to measure the independence between the prompts of two domains:

$$\text{HSIC}(\mathcal{D}_a, \mathcal{D}_b) = \frac{1}{(\rho - 1)^2} \text{tr}(KHLH), \quad (1)$$

where $H$ is the centering matrix $H_\rho = I_\rho - \frac{1}{\rho}\mathbf{1}\mathbf{1}^{\mathbf{T}}$, $K_{ij} = \phi(\mathcal{D}_{a_i}, \mathcal{D}_{a_j})$, $L_{ij} = \psi(\mathcal{D}_{b_i}, \mathcal{D}_{b_j})$, $\mathcal{D}_{a_i} \in \mathbb{R}^{1 \times d_p}$, $\phi$ and $\psi$ denote the kernel functions. HSIC = 0 indicates independence, when $\phi$ and $\psi$ are universal kernels. However, HSIC is not invariant to isotropic scaling, which can be addressed by normalizing HSIC which is known as Centered Kernal Alignment (CKA) (Nguyen et al., 2020; Raghu et al., 2021; Chen et al., 2023):

$$\text{CKA}(\mathcal{D}_a, \mathcal{D}_b) = \frac{\text{HSIC}(\mathcal{D}_a, \mathcal{D}_b)}{\sqrt{\text{HSIC}(\mathcal{D}_a, \mathcal{D}_a)\text{HSIC}(\mathcal{D}_b, \mathcal{D}_b)}}, \quad (2)$$

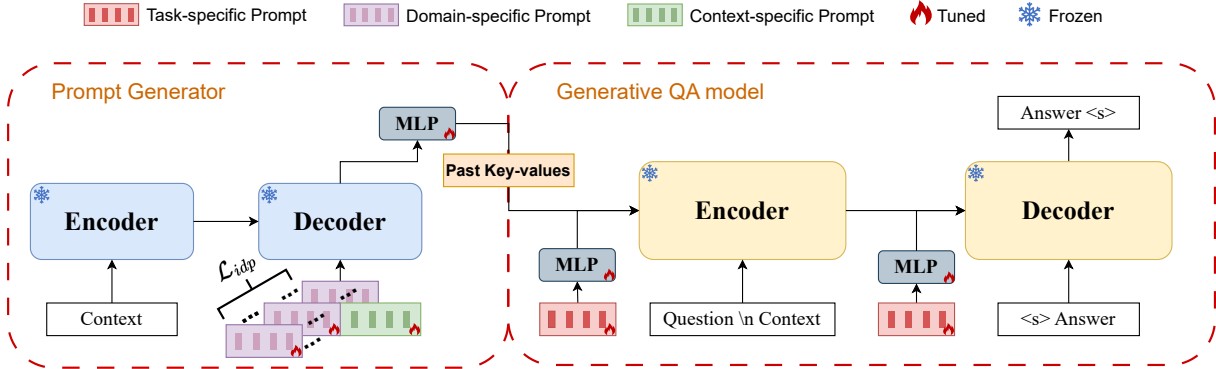

Figure 1: The overall framework of MPrompt.

where $\mathrm{CKA} \in [0, 1]$, and $\mathrm{CKA} = 0$ implies independence.

Computing the pair-wise independence requires $\frac{n(n-1)}{2}$ iterations, which is slow for large $n$. To reduce computational costs, we randomly sample $m$ pairs of domains as $\Theta$ to calculate the $\mathcal{L}_{idp}$ constraints in each training iteration:

$$\mathcal{L}_{idp} = \sum_{(i,j)\in\Theta} \mathrm{CKA}(\mathcal{D}_i, \mathcal{D}_j). \tag{3}$$

### 3.3 Context-specific Prompt

The domain-specific prompts provide shared intra-domain information, which provides fine-grained knowledge compared to task-specific prompts. However, there are still diversities among contexts within the same domain, and utilizing such diverse information is critical for answering questions accurately.

Therefore, we construct context-specific prompts to enhance the understanding of each context, which provides fine-grained knowledge compared to domain-specific prompts. Specifically, all contexts have a shared context-specific prompt $\mathcal{C} \in \mathbb{R}^{\kappa \times d_p}$, where $\kappa$ denotes the length of the context-specific prompt. Furthermore, we propose the prompt generator to ensure that $\mathcal{C}$ generates different prompts for different contexts, especially for those contexts unseen in the training data and discuss its other roles in the next section.

### 3.4 Prompt Generator

In general, task-specific prompts are related to the task of specific datasets, while domain-specific and context-specific prompts both are closely related to the context. To better leverage domain-specific and context-specific prompts to enhance PLMs' understanding of the context semantics, we introduce a small-scale PLM to encode contexts and integrate them into the prompt generation process.

For a context $c_i$, which belongs to the domain $D_j$. The encoder of the prompt generator takes the context $c_i$ as its input, while the concatenation of domain-specific prompt $\mathcal{D}_j$ and context-specific prompt $\mathcal{C}$ serves as the input $\mathcal{X}$ for the decoder,

$$\mathcal{X} = [\mathcal{D}_j; \mathcal{C}], \tag{4}$$

where $\mathcal{X} \in \mathbb{R}^{(\rho+\kappa)\times d_p}$. It should be noted that we have removed the original decoder embedding layer. The output of the prompt generator is mapped to key-value pairs $\mathcal{P} = \{\mathcal{P}_1, ..., \mathcal{P}_L\}$ through the MLP,

$$\mathcal{P} = \mathrm{MLP}(\mathrm{PromptGenerator}(c_i, \mathcal{X})), \tag{5}$$

where $\mathcal{P} \in \mathbb{R}^{(\rho+\kappa)\times 2dL}$, $\mathcal{P}_l = (\mathcal{P}_{l,K}, \mathcal{P}_{l,V})$, $\mathcal{P}_{l,K}$ and $\mathcal{P}_{l,V} \in \mathbb{R}^{(\rho+\kappa)\times d}$, and $L$ denotes the number of layers in the generative QA model. Intuitively, the knowledge related to the context $c_i$ is steered from the encoder of PLMs, and then integrated into the prompt generation process in the decoder. In this way, our approach allows for better learning of the semantics between prompt and context than previous work (Li and Liang, 2021; Lester et al., 2021; Ma et al., 2022), since both domain-specific prompt and context-specific prompt are closely related to the context.

### 3.5 Applying Multi-level Prompts

Overall, $\mathcal{P}$ contains the information of domain-specific and context-specific prompts as well as knowledge from PLMs related to the context, while $\mathcal{T}_{task}$ contains the shared information within the task. In order to exploit multi-level prompt information to enhance the performance on question answering, we integrate the above different levels

of prompts into the encoder of the generative QA model. Specifically, for the self-attention computation of layer $l$ in the encoder of the generative QA model, the original $K_l$ and $V_l$ are augmented as:

$$K_l' = [\mathcal{T}_{E_{l,K}}; \mathcal{P}_{l,K}; K_l],$$
$$V_l' = [\mathcal{T}_{E_{l,V}}; \mathcal{P}_{l,V}; V_l] \tag{6}$$

where $K_l'$ and $V_l' \in \mathbb{R}^{(t+\rho+\kappa+M)\times d}$, $M$ denotes the length of the input sequence. For the self-attention and cross-attention computation of layer $l$ in the decoder, $K_l$ and $V_l$ are augmented as:

$$K_l' = [\mathcal{T}_{Dm(Dc)_{l,K}}; K_l], V_l' = [\mathcal{T}_{Dm(Dc)_{l,V}}; V_l] \tag{7}$$

where $K_l'$ and $V_l' \in \mathbb{R}^{(t+M)\times d}$.

To train the multi-level prompts, the loss function is a weighted sum of the two loss terms:

$$\mathcal{L} = \mathcal{L}_{\text{NLL}} + \lambda \mathcal{L}_{idp}, \tag{8}$$

where $\lambda$ is the hyperparameter used to control the independence constraint, $\mathcal{L}_{\text{NLL}}$ is the text generation loss, as follows:

$$\mathcal{L}_{\text{NLL}} = -\sum_{t=1}^{N} \log p(y_t | x, y_{<t}), \tag{9}$$

where $y_t$ denotes the $t$-th element of the target sequence, and $x$ represents the input sequence. It is worth noting that, guided by Equation 8, we only update the MLP, task-specific, domain-specific, and context-specific prompts, while keeping all other parameters frozen.

## 4 Experiments

### 4.1 Datasets and Baselines

**Datasets.** To cover a wide range of QA tasks in our experiments, we evaluated our approach on 12 benchmark datasets in the fields of **Extractive QA (EX)**: SQuAD2 (Rajpurkar et al., 2018), NewsQA (Trischler et al., 2016), **Abstractive QA (AB)**: NarrativeQA (Kočiskỳ et al., 2018), DROP (Dua et al., 2019), **Multiple-choice QA (MC)**: MCTest (Richardson et al., 2013), ARC(easy, challenge) (Clark et al., 2016, 2018), OpenBookQA (Mihaylov et al., 2018), QASC (Khot et al., 2020),RACE (Lai et al., 2017), and **Yes/No QA (YN)**: BoolQ (Clark et al., 2019), BoolQ-NP (Khashabi et al., 2020a). Table 1 presents the statistics of these datasets. Following Khashabi et al. (2020b), the above-mentioned

| Datasets | #Train | #Eval. | #Test | Ques. len. | Cont. len. | Ans. len. | Type |
|---|---|---|---|---|---|---|---|
| SQuAD2 | 118446 | 11873 | 11873 | 9.8 | 120 | 2.7 | EX |
| NewsQA | 72219 | 4341 | 4341 | 6.6 | 611 | 4.1 | EX |
| NarQA | 65494 | 6922 | 21114 | 8.5 | 572 | 4.1 | AB |
| DROP | 67864 | 9536 | 9536 | 10.7 | 207 | 1.5 | AB |
| MCTest | 1480 | 320 | 840 | 26.2 | 213 | 3.9 | MC (4) |
| ARC(easy) | 2250 | 569 | 2367 | 39.1 | 189 | 3.8 | MC (4) |
| ARC(chal.) | 1119 | 299 | 1172 | 46.2 | 185 | 4.9 | MC (4) |
| OBQA | 4957 | 500 | 500 | 26.8 | 155 | 2.9 | MC (4) |
| QASC | 7208 | 926 | 926 | 30.1 | 253 | 1.6 | MC (8) |
| RACE | 25421 | 1436 | 1436 | 33.3 | 191 | 4.6 | MC (4) |
| BoolQ | 6157 | 3270 | 3270 | 8.8 | 96 | 1.0 | YN |
| BoolQ-NP | 9727 | 3798 | 3798 | 9.1 | 98 | 1.0 | YN |

Table 1: Dataset Statistics. NarQA and OBQA refer to NarrativeQA and OpenBookQA. "#Train" is an abbreviation for "the number of Training set". "Ques.", "Cont.", "Ans.", and "len." are abbreviations for "Question", "Context", "Answer", and "Length", respectively. MC(4) indicates that the dataset contains 4 candidates.

datasets in different formats were converted to a unified format to suit generative QA tasks. Due to space limitations, more details are available in Appendix A.1.

**Metrics.** We evaluate each dataset using the metrics most often used in previous work. For SQuAD2 and DROP, we used the F1 score with token overlap between the answer text and the gold answers. For NewsQA and NarrativeQA, we use ROUGE-L metric (Lin, 2004). For the multiple-choice and Yes/No QA, we use accuracy for evaluation (sometimes referred to as exact match), i.e., a generated answer is considered correct only if it exactly matches the gold answers.

**Baselines.** To comprehensively evaluate the performance of MPrompt, we compared it with a wide range of state-of-the-art soft-prompt methods, such as Fine-tuning (Khashabi et al., 2022), Prefix-tuning (Li and Liang, 2021), Prompt-tuning (Lester et al., 2021) and XPrompt (Ma et al., 2022).

### 4.2 Implementation

We convert each dataset into a unified text-to-text format to suit generative question answering models following (Khashabi et al., 2020b, 2022). Our MPrompt is based on three scales of pre-trained UnifiedQA (Khashabi et al., 2020b) (which is a T5 model for question-answering tasks): Base, Large, XL with 220M, 770M and 3B parameters, respectively. For the prompt generator, we utilize UnifiedQA-Small with 60M parameters to ensure that there is no excessive demand for GPU memory.

In all experiments, we employ the AdamW optimizer (Loshchilov and Hutter, 2017) and set $\beta_1 = 0.9$, $\beta_2 = 0.999$, and the weight decay is

0.01. We train our method with a learning rate of 5e-5, 10% warmup ratio, $\lambda$=1e-4, 50 epochs and record the model with the best performance on the validation set. To ensure a fair comparison, we fix the length of task-specific prompts to 10 and adjust the lengths of domain-specific and context-specific prompts to {5, 10, 15, 20, 30, 40, 50, 60}. We use Kmeans (MacQueen, 1967) and Sentence-Transformers (all-mpnet-base-v2) (Reimers and Gurevych, 2019) to cluster the context and fix the number of clusters to 3 to obtain domain information D. The visualization of the clustering results by t-SNE (Van der Maaten and Hinton, 2008) is deferred to Appendix A.2. For all baselines, all hyperparameter settings are based on the reported values in the original paper to achieve optimal results. Our method is implemented with PyTorch (Paszke et al., 2019) and Transformers (Wolf et al., 2020) library and experiments are conducted on Ubuntu 22.04 systems with NVIDIA RTX A100 or 4090 GPUs. Other implementation details and optimal hyperparameters are deferred to Appendix A.3.

### 4.3 Performance Comparison

Table 2 displays the main experimental results of different methods on 12 benchmark datasets. We conduct a comprehensive comparison between MPrompt and state-of-the-art methods, including Prompt-tuning (Lester et al., 2021), Prefix-tuning (Li and Liang, 2021), and XPrompt (Ma et al., 2022) for different parameter sizes of PLMs. The datasets cover a wide range of question-answering scenarios, which is beneficial for the comprehensive evaluation of different methods.

We observe that: (1) Our method MPrompt outperforms other soft-prompt methods by a large margin across all tasks and model scales. For example, MPrompt achieves absolute improvements of 2.17%, 1.85%, and 1.82% relative to Prefix-tuning on UnifiedQA-Base, Large, and XL respectively. It is due to the input-independent prompt learning methods applying a uniform prompt to all inputs for a given task, which evidently underutilizing the input semantics in answer generation. However, MPrompt significantly improves the performance in question-answering tasks by enhancing the contextual comprehension of the PLMs with multiple levels of prompts. (2) Prefix-tuning and XPrompt have comparable performance at the same model size. Both algorithms outperform Prompt-tuning on the NewsQA, DROP,

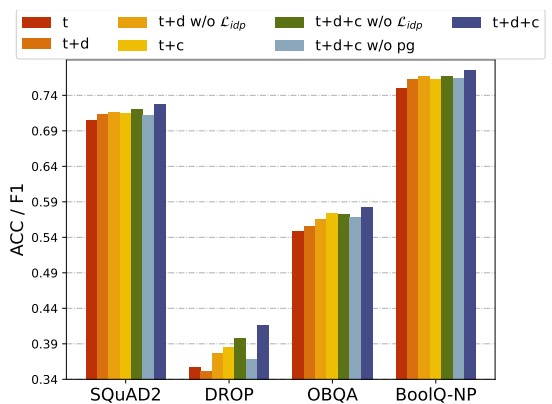

Figure 2: Ablation Study. "t", "d", and "c" denote task-specific, domain-specific, and context-specific prompts, respectively. "w/o" is an abbreviation for "without".

OBQA, QASC, and BoolQ-NP datasets. It is because Prefix-tuning provides deeper prompts, while XPrompt removes negative prompts in Prompt-tuning. However, MPrompt achieves higher performance than Prefix-tuning and XPrompt at the same model sizes, demonstrating its effectiveness. (3) Due to the luxury of having high computational resources and a full-weight update scheme in full fine-tuning, there is still a significant performance gap between soft-prompt tuning and full fine-tuning. However, As shown in Table 2, MPrompt matches the fine-tuning performance on all tasks and even outperforms the fine-tuning performance of UnifiedQA-Base and XL on most tasks. Specifically for UnifiedQA-Base, MPrompt achieves the best performance on SQuAD2, NewsQA, NarQA, MCTest, ARC (easy), RACE, and BoolQ, resulting in +0.69%, +0.62%, +0.24%, +1.31%, +0.78%, 0.21%, and 0.25% improvements over fine-tuning, respectively. We incorporate context knowledge from other PLMs (such as UnifiedQA-small in this paper) into prompt generation to enrich the semantics.

In summary, our method achieved excellent performance compared to state-of-the-art soft prompt methods, closing and even surpassing the performance gap over fine-tuning. This demonstrates that MPrompt effectively enhances contextual comprehension and enriches the semantics of the PLMs which significantly improves the quality of downstream question-answering tasks.

### 4.4 Ablation Analysis

In this part, we perform an ablation study on the various components of MPrompt, as shown in Figure

| | Model | SQuAD2 F1 | NewsQA ROUGE-L | NarQA ROUGE-L | DROP F1 | MCTest ACC | ARC(easy) ACC | ARC(chall.) ACC | OBQA ACC | QASC ACC | RACE ACC | BoolQ ACC | BoolQ-NP ACC |
|---|---|---|---|---|---|---|---|---|---|---|---|---|---|
| | Fine-tuning | 71.92 | 59.36 | 46.06 | 43.50 | 86.43 | 72.45 | 45.31 | 58.60 | 69.55 | 75.49 | 82.72 | 78.52 |
| | Prompt-tuning | 68.07 | 54.83 | 44.96 | 25.99 | 85.36 | 68.86 | 42.11 | 45.60 | 56.16 | 72.98 | 82.23 | 72.59 |
| Base | Prefix-tuning | 71.45 | 57.70 | 45.23 | 35.72 | 85.95 | 70.68 | 42.49 | 55.00 | 68.17 | 73.51 | 82.32 | 76.31 |
| 220M | XPrompt | 70.49 | 57.87 | 45.15 | 31.32 | 85.75 | 71.56 | 42.73 | 53.20 | 64.47 | 73.73 | 82.45 | 75.48 |
| | **MPrompt** | **72.61** | **59.99** | **46.30** | **41.64** | **87.74** | **73.23** | **44.97** | **58.20** | **69.98** | **75.70** | **82.97** | **77.25** |
| | Improvement | ↑ 1.16 | ↑ 2.29 | ↑ 1.07 | ↑ 5.92 | ↑ 1.79 | ↑ 2.56 | ↑ 2.47 | ↑ 3.20 | ↑ 1.81 | ↑ 2.19 | ↑ 0.64 | ↑ 0.94 |
| | Fine-tuning | 78.13 | 59.77 | 50.20 | 52.20 | 91.67 | 81.23 | 54.95 | 67.40 | 80.35 | 81.48 | 86.39 | 84.36 |
| | Prompt-tuning | 72.40 | 56.45 | 48.76 | 38.88 | 90.24 | 76.47 | 52.05 | 56.00 | 61.77 | 78.43 | 85.04 | 79.27 |
| Large | Prefix-tuning | 75.20 | 59.24 | 48.21 | 43.79 | 92.20 | 79.71 | 52.67 | 64.60 | 78.08 | 79.50 | 85.45 | 81.83 |
| 770M | XPrompt | 75.54 | 58.16 | 48.56 | 42.04 | 92.28 | 78.28 | 53.13 | 61.20 | 73.91 | 80.18 | 85.83 | 81.95 |
| | **MPrompt** | **76.52** | **60.35** | **49.37** | **50.08** | **93.45** | **80.93** | **54.50** | **67.00** | **80.15** | **81.19** | **86.17** | **82.94** |
| | Improvement | ↑ 1.32 | ↑ 1.11 | ↑ 1.16 | ↑ 6.29 | ↑ 1.26 | ↑ 1.22 | ↑ 1.83 | ↑ 2.40 | ↑ 2.07 | ↑ 1.69 | ↑ 0.72 | ↑ 1.10 |
| | Fine-tuning | 87.66 | 64.54 | 65.85 | 62.98 | 95.71 | 86.76 | 66.54 | 80.60 | 89.95 | 85.26 | 89.38 | 87.70 |
| | Prompt-tuning | 82.91 | 60.24 | 53.69 | 45.36 | 93.33 | 83.75 | 63.40 | 71.00 | 85.54 | 85.28 | 88.39 | 84.65 |
| XL | Prefix-tuning | 84.86 | 61.83 | 57.34 | 58.83 | 95.27 | 85.77 | 66.98 | 78.20 | 86.01 | 85.44 | 88.93 | 86.26 |
| 3B | XPrompt | 84.73 | 62.24 | 56.66 | 51.11 | 94.39 | 85.98 | 65.69 | 76.30 | 86.87 | 85.91 | 89.47 | 86.49 |
| | **MPrompt** | **86.48** | **63.37** | **59.41** | **60.02** | **96.43** | **86.95** | **69.71** | **81.80** | **88.98** | **86.71** | **90.27** | **87.39** |
| | Improvement | ↑ 1.62 | ↑ 1.54 | ↑ 2.07 | ↑ 1.19 | ↑ 1.16 | ↑ 1.18 | ↑ 2.73 | ↑ 3.60 | ↑ 2.97 | ↑ 1.27 | ↑ 1.34 | ↑ 1.13 |

Table 2: Comparison of state-of-art algorithm on different datasets. The unit for all the metrics here is in percentage(%). The numbers in blue indicate the performance gain (↑) of our method compared to Prefix-tuning.

2. Firstly, we observe a decrease in performance when removing domain-specific or context-specific prompts. The domain-specific or context-specific prompts are constructed based on inputs of different granularity, which enhances the semantic comprehension of the input. Secondly, when removing the independence constraint, there was a significant decrease in performance. The independence constraint steers domain-specific prompts to focus on intra-domain information rather than inter-domain information, which can effectively avoid information redundancy. Furthermore, performance decreases when the prompt generator is removed. The prompt generator ensures that context-specific prompts are generated differently for different contexts, even those that never appear in the training data, which enhances the semantic understanding of the input context. Moreover, the prompt generator elicits context-related knowledge from PLM and incorporates it into the prompt generation process, which helps improve the context awareness of the prompts.

### 4.5 Sensitivity Analyses

In this part, we conducted comprehensive sensitivity analyses on our proposed method, including the length of prompts, the weight $\lambda$ of the loss $\mathcal{L}_{idp}$, different clustering results D, different scales of PLMs in the prompt generator, and the number of sampled domain pairs $m$.

#### 4.5.1 The Length of Prompts

In MPrompt, the length of prompts is a key factor that affects model performance. Here, we investigate how the length of domain-specific and

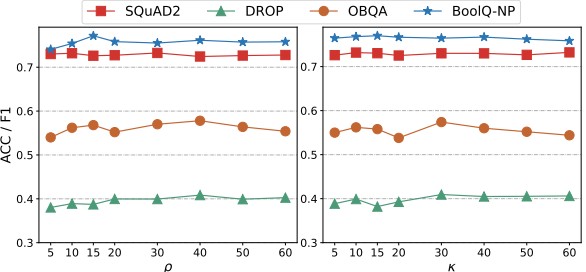

Figure 3: Evaluation of prompt length on the UnifedQA-base model. $\rho$ and $\kappa$ denote the length of the domain-specific and context-specific prompts, respectively.

context-specific prompts impacts the final performance. We fixed the length of one prompt to 10 and varied the other in the range of {5, 10, 15, 20, 30, 40, 50, 60}. As shown in Figure 3, in most cases, MPrompt shows stable performance for the length of domain-specific and context-specific prompts. Moreover, since DROP and OBQA require reasoning ability (Roberts et al., 2020), they are more sensitive to the prompt length compared to other datasets.

#### 4.5.2 The Weight of Loss $\mathcal{L}_{idp}$

We investigated the impact of loss weighing $\lambda$ on the results, as shown in Table 3. We found the change of weighting has minor impact on the SQuAD2 dataset and there is an optimal weight of 0.0001 for DROP, OBQA, and BoolQ-NP datasets. $\mathcal{L}_{idp}$ takes values between $[0, 1]$, a too large $\lambda$ means that the model is not focusing on generating answers as its primary goal. An extremely small $\lambda$ would make the domain-specific prompts lose focus on unique intra-domain information.

| $\lambda$ | SQuAD2 | DROP | OBQA | BoolQ-NP |
|---|---|---|---|---|
| 1 | 72.60 | 38.72 | 55.80 | 76.28 |
| 0.1 | **72.67** | 39.12 | 55.60 | 77.15 |
| 0.01 | 72.61 | 40.87 | 57.80 | 74.93 |
| 0.001 | 72.63 | 41.21 | 58.20 | 76.38 |
| 0.0001 | 72.61 | **41.67** | **58.60** | **77.25** |
| 0.00001 | 72.63 | 39.20 | 56.00 | 76.25 |

Table 3: Evaluation of $\lambda$ on UnifiedQA-base model.

### 4.5.3 Clustering Results

We investigated the impact of different numbers of clusters on performance, as shown in Table 4. Since the gold label of clustering results is not available in the question-answering datasets, it is difficult to determine the optimal number of clusters. Our evaluation shows, the performance of the model is not sensitive to the number of clusters. KMeans always outperforms randomly assigning cluster labels, which demonstrates that introducing contextual cluster information to the model improves context comprehension.

| | # clusters | SQuAD2 | DROP | OBQA | BoolQ-NP |
|---|---|---|---|---|---|
| | 3 | 71.37 | 37.26 | 56.20 | 75.59 |
| Random | 6 | 71.63 | 37.25 | 55.40 | 75.91 |
| | 9 | 71.62 | 37.28 | 56.80 | 75.25 |
| | 3 | 72.61 | 41.67 | 58.20 | 77.25 |
| KMeans | 6 | 72.70 | 40.71 | 58.60 | 76.93 |
| | 9 | 72.71 | 40.93 | 57.90 | 76.01 |

Table 4: Evaluation of the number cluster on the UnifiedQA-base model. "Random" refers to randomly assigning cluster labels.

### 4.5.4 Different Scales of Prompt Generator

In general, increasing the parameter number of PLMs brings abundant semantic knowledge. Therefore, we investigated the impact of PLMs with different scales on performance, as shown in Figure 4. The prompt generator delivers significant performance improvements. Our evaluation shows, larger-scale PLMs tend to have better results, but require more computational resources. To balance the trade-off between cost and performance, the UnifiedQA-small already delivers satisfactory performance gains with a small computational overhead (60M parameters).

### 4.5.5 Number of sampled domain pairs

We investigated the impact of sampled domain pairs on the results. The number of clusters is set to 6, which requires 15 iterations per batch. We evaluate the number of sample pair $m$ in {1, 3, 5,

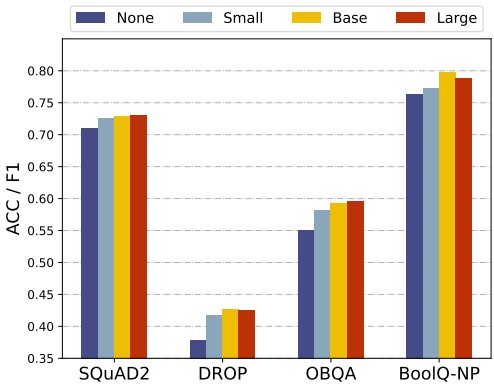

Figure 4: Experiment of the prompt generator on different UnifiedQA models. "None" indicates that domain-specific and context-specific prompts were trained in the same way as task-specific prompts.

10, 15}. Our evaluation in Table 5 shows that our algorithm is not sensitive to the number of sampled domain pairs $m$. Even with a smaller $m$ per batch, it still provides sufficient sampling frequency in training, which greatly reduces the computational costs.

| $m$ | SQuAD2 | DROP | OBQA | BoolQ-NP |
|---|---|---|---|---|
| 1 | 72.52 | 38.99 | 57.40 | 76.47 |
| 3 | 72.70 | 40.71 | 58.60 | 76.93 |
| 5 | 72.75 | 40.01 | 58.80 | 76.88 |
| 10 | 72.65 | 40.11 | 59.00 | 77.17 |
| 15 | 72.76 | 41.30 | 58.60 | 76.96 |

Table 5: Evaluation of different the number of sampled domain pairs per batch $m$.

## 5 Conclusion

In this paper, we propose a novel **M**ulti-level **Prompt** (MPrompt) tuning method for machine reading comprehension. Our method strengthens PLMs' utilization of input semantics through three levels of prompts: task-specific prompts, domain-specific prompts, and context-specific prompts. The task-specific prompts are input-independent and generate prompts specific to a task. The domain-specific prompts utilize the domain knowledge generated from the dataset while context-specific prompts are relying on the input context. Our experiments show the combination of three level prompts improves the answer generation performance on different sizes of PLMs and 12 benchmark datasets. In future work, we will extend our method to more tasks such as summarization, translation, and sentiment analysis.

## Limitations

In our method, the length of prompts is the most critical parameter that affects performance. In our experiments, we observe that MPrompt is sensitive to prompt length for some challenging datasets. To obtain the optimal hyperparameter combination, it is inevitable to perform a grid search on the length of prompts. Our model is designed for encoder-decoder structure, so the decoder-only structure like LLaMA, GPT, or Bloom is not applicable. Our model requires access to the parameter of the model which any black box model is not applicable to our algorithm.

## Ethics Statement

Our work is developed with the highest ethical standards in mind. Our work should not be used for any entity that may violate human rights.

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

# A Appendix

## A.1 Datasets: Details

We evaluated our method on 12 datasets covering a wide range of QA tasks. Due to some datasets (such as ARC, OpenBookQA and QASC) lacking the context, following Khashabi et al. (2020b, 2022), we used the datasets that contain retrieved contexts. Due to limited test access for some datasets, such as SQuAD2, NewsQA, DROP, QASC, BoolQ, and BoolQ-NP, we used the validation set as the test set and re-randomized an equal number of samples from the training set as the validation set. For MCTest, we used the sum of mc160 and mc500. For RACE, we used RACE-middle, which consists of English reading comprehension questions designed for Chinese middle school students. The datasets would be available in our code.

## A.2 Visualization of context clustering results with Kmeans

In the paper, we cluster the contexts by Kmeans and fix the number of clusters to 3, since we do not have access to the gold standard clustering results for each dataset. To observe the results of clustering, we conducte visualization using t-SNE (Van der Maaten and Hinton, 2008), as shown in Figure 5. Most of the datasets present better clustering results when the number of clusters is 3, which will provide better domain information.

## A.3 Implementation details

In Table 6, we report the hyperparameters used for training our models recorded in the experimental section. For model inference (answer generation), we set num_beams to 2, min_length to 1, and early_stopping to True. For MLP, we set the hidden layer dimension to 512 and utilize the Tanh activation function. For domain-specific prompts and context-specific prompts, we initialize each prompt token as an embedded vector extracted from the prompt generator's vocabulary, as Lester et al. (2021) done.

| Datasets | Task Len | Domain len | Context Len | dropout | bsz | max_ans_length |
|---|---|---|---|---|---|---|
| SQuAD2 | 10 | 10 | 60 | 0.1 | 16 | 150 |
| NewsQA | 10 | 20 | 60 | 0.1 | 16 | 250 |
| NarQA | 10 | 50 | 60 | 0 | 16 | 100 |
| DROP | 10 | 50 | 50 | 0.1 | 16 | 150 |
| MCTest | 10 | 10 | 30 | 0 | 5 | 170 |
| ARC(easy) | 10 | 20 | 15 | 0 | 5 | 50 |
| ARC(chal.) | 10 | 40 | 40 | 0 | 5 | 80 |
| OBQA | 10 | 15 | 15 | 0 | 5 | 50 |
| QASC | 10 | 50 | 30 | 0.1 | 10 | 80 |
| RACE | 10 | 20 | 5 | 0.1 | 15 | 70 |
| BoolQ | 10 | 30 | 60 | 0.1 | 8 | 10 |
| BoolQ-NP | 10 | 15 | 10 | 0 | 10 | 10 |

Table 6: Hyperparameter settings for our method. "Task len" indicates the token length of task-specific prompts. "bsz" indicates batch size. "max_ans_length" indicates the maximum length of generated answers during inference.

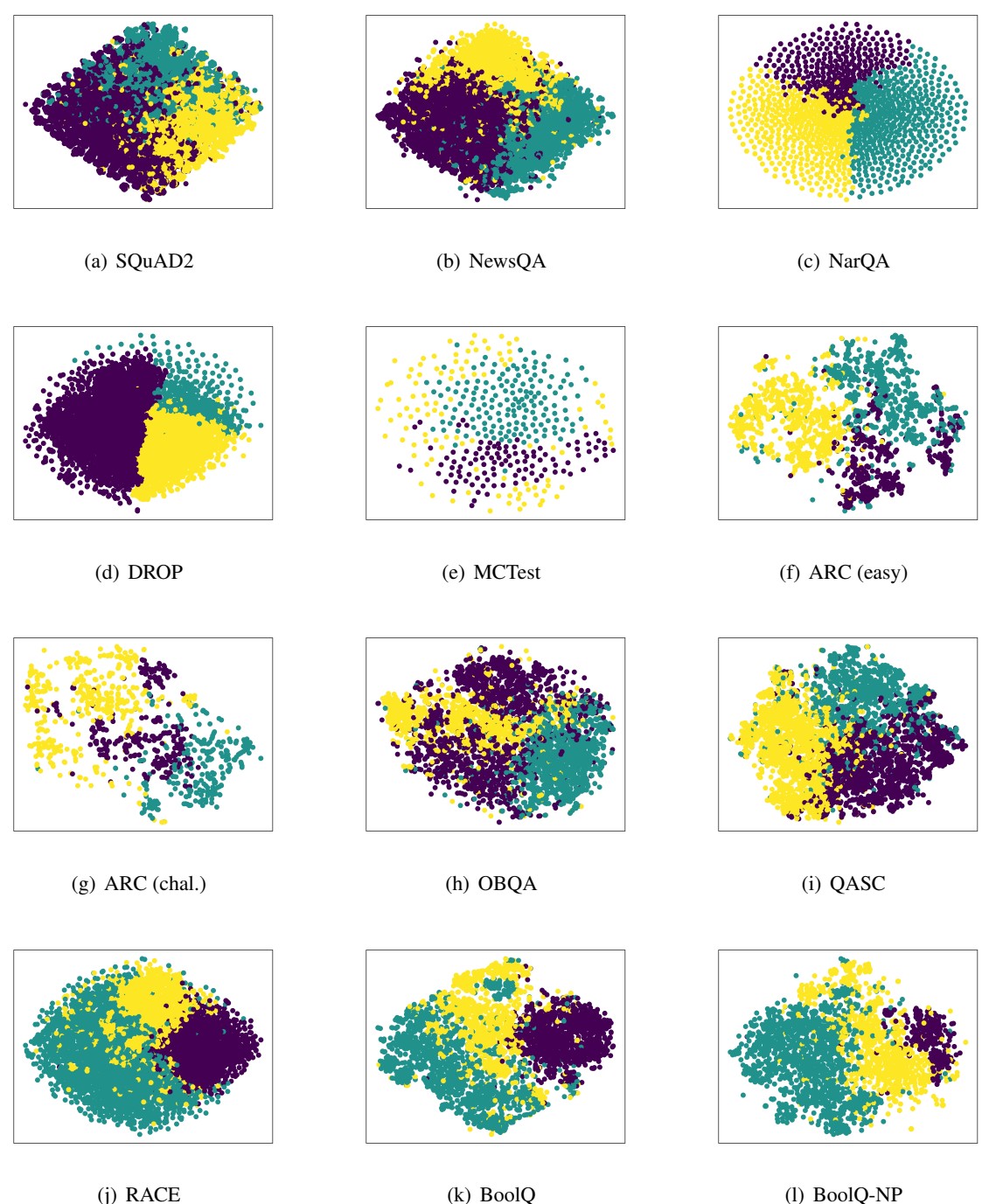

Figure 5: The visualization of Kmeans clustering results for context by t-SNE.