# OpenReview forum: "MPrompt: Exploring Multi-level Prompt Tuning for Machine Reading Comprehension"
_EMNLP/2023/Conference — EMNLP 2023 Findings_

### Official Review · Reviewer_Pw7P · 2023-08-02

**Soundness:** 3

**Excitement:**

3: Ambivalent: It has merits (e.g., it reports state-of-the-art results, the idea is nice), but there are key weaknesses (e.g., it describes incremental work), and it can significantly benefit from another round of revision. However, I won't object to accepting it if my co-reviewers champion it.

**Paper Topic And Main Contributions:**

This paper introduces Multi-level Prompt Tuning (MPrompt), for
parameter efficient learning of the reading comprehension (MRC)
task. In addition to the task prompt vectors used in prefix/prompt
tuning, MPrompt proposes to use extra prompt vectors that are dynamically
created (generated by a small LM) for different domains and contexts
in the reading comprehension task.

To decide the domains, an MRC data will be firstly clustered with
Kmeans to decide the number of domains; for each of the identified
domain, prompt vectors are generated from an LM. Furthermore, for each
of the contexts in MRC, extra prompt vectors are also generated from
an LM. Authors argue that this multi-level prompt tuning better learns
the MRC task, and show performance improvements on MRC datasets over
previous methods including prefix/prompt tuning and XPrompt.

**Questions For The Authors:**

QuestionA: line235: I wonder how does the trained model generalize to
new domains that are not in the training data? How should we decide
which group of domain prompts should we use for the new domains?

QuestionB: line450: MPrompt outperforms the baselines. But I wonder
how efficient is MPrompt, when comparing with the baselines?  Does
MPrompt use more trainable parameters? In general, I think a detailed
comparison on the efficiency of different methods would be very
helpful.

**Reasons To Accept:**

- The proposed method MPrompt is intuitive and shows improvements over
  several MRC/QA datasets;

- The paper is well-structured and easy to follow.

**Reasons To Reject:**

I have three major concerns:

- I have some concerns on the *generalization ability* of MPrompt.
  Firstly, MPrompt relies on clustering of the MRC dataset to determine
  the domains. So I think it is difficult for MPrompt to solve new (or
  OOD) examples. We can allocate the new example to the closest
  cluster, but the performance is unknown and may have large
  variances. Second, MPrompt is specifically tailored for the reading
  comprehension task, and it seems not easy to be adapted to other
  scenarios where efficient NLP methods are needed.

- *Comparison with baselines.* MPrompt outperforms prefix/prompt tuning
  and XPrompt, however, the extra cost is unknown. For example, I
  think MPrompt introduces more trainable parameters than method like
  prompt tuning; an extra LM is also leveraged to generate the
  prompts. I think it is beneficial to have some detailed comparisons
  on the cost or efficiency of these methods, which is crucial
  for parameter-efficient training.

- As described in this paper, generating instance-level prompts has
  been shown to be more effective than using task-level prompts alone.
  So MPrompt seems a direct adaptation of instance-level prompts to the machine reading
  comprehension task, so I feel the novelty is limited.

**Reproducibility:**

4: Could mostly reproduce the results, but there may be some variation because of sample variance or minor variations in their interpretation of the protocol or method.

**Reviewer Confidence:**

4: Quite sure. I tried to check the important points carefully. It's unlikely, though conceivable, that I missed something that should affect my ratings.

---

> ### Author Rebuttal · Authors · 2023-08-28
>
> Dear Reviewer,
>
> We would like to express our gratitude for your insightful comments and suggestions. Your feedback has allowed us to improve the quality of our study significantly. Please find our responses to your queries and concerns below.
>
> ## For your concerns on the generalization ability of MPrompt (as well as QuestionA).
> Generalization capability primarily refers to the performance when encountering new datasets, particularly in zero-shot and few-shot scenarios, as MPrompt can be retrained under fully supervised scenarios.
> Thus, we evaluate the performance of our method (MPrompt) in zero-shot and few-shot scenarios across 6 MC datasets and compare it with Prefix-tuning, as shown in Table 1~6.
>
> ### In the zero-shot scenarios.
> Table 1: The zero-shot performance of **MPrompt** across 6 MC datasets. 'In-domain' refers to the best results achieved through training and testing on the same dataset.
> ____
> | Trained on $\downarrow$ - Evaluated on $\rightarrow$ | MCTest | ARC(easy) | ARC(chall.) | OBQA   | QASC   | RACE   |
> |:----------------------------------------------------:|:------:|:---------:|:-----------:|:------:|:------:|:------:|
> | MCTest                                               | n/a    | 66.67     | 42.49       | 32.80  | 48.06  | 73.40  |
> | ARC(easy)                                            | 77.26  | n/a       | 41.13       | 41.00  | 44.38  | 65.74  |
> | ARC(chall.)                                          | 66.31  | 57.37     | n/a         | 50.00  | 34.34  | 60.79  |
> | OBQA                                                 | 40.12  | 47.47     | 41.13       | n/a    | 20.30  | 40.53  |
> | QASC                                                 | 70.60  | 69.61     | 35.32       | 27.40  | n/a    | 60.52  |
> | RACE                                                 | 85.71  | 67.63     | 43.09       | 34.40  | 50.54  | n/a    |
> | In-domain                                            | 87.74  | 73.23     | 44.97       | 58.20  | 70.95  | 75.70  |
>
> Table 2: The zero-shot performance of **Prefix-tuning** across 6 MC datasets.
> ____
> | Trained on $\downarrow$ - Evaluated on $\rightarrow$ | MCTest | ARC(easy) | ARC(chall.) | OBQA   | QASC   | RACE   |
> |:----------------------------------------------------:|:------:|:---------:|:-----------:|:------:|:------:|:------:|
> | MCTest                                               | n/a    | 61.85     | 39.75       | 32.20  | 48.81  | 73.05  |
> | ARC(easy)                                            | 75.00  | n/a       | 40.02       | 32.60  | 44.95  | 62.28  |
> | ARC(chall.)                                          | 72.14  | 53.62     | n/a         | 48.40  | 31.53  | 59.19  |
> | OBQA                                                 | 40.45  | 49.58     | 40.10       | n/a    | 22.79  | 43.31  |
> | QASC                                                 | 69.17  | 70.50     | 36.95       | 25.60  | n/a    | 66.57  |
> | RACE                                                 | 83.60  | 67.89     | 40.80       | 36.40  | 49.46  | n/a    |
>
> As evidenced in Tables 1 and 2, MPrompt delivers comparable performance to Prefix-tuning in zero-shot scenarios, even achieving a 0.59% improvement over Prefix-tuning.
> We contend that the prompt generation process for domain-specific prompts, which incorporates contextual knowledge encoded from a small PLM, significantly reduces variance. This can also be supported by the ablation study in our paper (please refer to Figure 2 in our paper, the prompt generator has a key role to play in performance improvement).
>
> ### In the few-shot scenarios.
> Table 3: The performance of **MPrompt** in few-shot (1%) scenarios across 6 MC datasets. '$\uparrow$' denotes the average improvement compared to zero-shot scenarios.
> _____
> | Trained on $\downarrow$ - Evaluated on $\rightarrow$ | MCTest | ARC(easy) | ARC(chall.) | OBQA   | QASC   | RACE   | $\uparrow$ |
> |:----------------------------------------------------:|:------:|:---------:|:-----------:|:------:|:------:|:------:|:----------:|
> | MCTest                                               | n/a    | 66.50     | 42.66       | 41.00  | 48.81  | 73.75  | 1.86%      |
> | ARC(easy)                                            | 80.00  | n/a       | 41.55       | 48.40  | 48.70  | 72.14  | 4.26%      |
> | ARC(chall.)                                          | 80.60  | 57.20     | n/a         | 51.60  | 44.38  | 72.01  | 7.39%      |
> | OBQA                                                 | 42.62  | 49.96     | 41.55       | n/a    | 42.12  | 72.77  | 11.89%     |
> | QASC                                                 | 76.90  | 70.71     | 37.54       | 38.20  | n/a    | 69.43  | 5.87%      |
> | RACE                                                 | 85.60  | 67.63     | 43.17       | 38.80  | 49.89  | n/a    | 0.74%      |
> | In-domain                                            | 87.74  | 73.23     | 44.97       | 58.20  | 70.95  | 75.70  |            |
>
> Table 4: The performance of **Prefix-tuning** in few-shot (1%) scenarios across 6 MC datasets.
> ___
> | Trained on $\downarrow$ - Evaluated on $\rightarrow$ | MCTest | ARC(easy) | ARC(chall.) | OBQA   | QASC   | RACE   |
> |:----------------------------------------------------:|:------:|:---------:|:-----------:|:------:|:------:|:------:|
> | MCTest                                               | n/a    | 63.51     | 42.75       | 37.20  | 48.86  | 73.40  |
> | ARC(easy)                                            | 78.24  | n/a       | 40.02       | 34.20  | 44.84  | 70.98  |
> | ARC(chall.)                                          | 77.26  | 54.00     | n/a         | 48.60  | 43.20  | 71.05  |
> | OBQA                                                 | 41.31  | 50.08     | 40.53       | n/a    | 32.07  | 72.28  |
> | QASC                                                 | 74.48  | 70.12     | 37.71       | 36.60  | n/a    | 67.17  |
> | RACE                                                 | 85.60  | 66.85     | 41.80       | 38.80  | 48.08  | n/a    |
>
>
> Table 5: The performance of **MPrompt** in few-shot (5%) scenarios across 6 MC datasets.
> _____
> | Trained on $\downarrow$ - Evaluated on $\rightarrow$ | MCTest | ARC(easy) | ARC(chall.) | OBQA   | QASC   | RACE   | $\uparrow$ |
> |:----------------------------------------------------:|:------:|:---------:|:-----------:|:------:|:------:|:------:|:----------:|
> | MCTest                                               | n/a    | 69.36     | 42.66       | 51.60  | 57.99  | 73.54  | 6.35%      |
> | ARC(easy)                                            | 84.40  | n/a       | 41.04       | 50.60  | 59.07  | 73.12  | 7.74%      |
> | ARC(chall.)                                          | 83.10  | 65.15     | n/a         | 53.20  | 57.45  | 72.91  | 12.60%     |
> | OBQA                                                 | 84.05  | 51.47     | 42.06       | n/a    | 56.05  | 73.40  | 23.50%     |
> | QASC                                                 | 80.12  | 73.19     | 37.71       | 49.80  | n/a    | 71.66  | 9.81%      |
> | RACE                                                 | 85.00  | 71.68     | 43.52       | 45.80  | 57.67  | n/a    | 4.46%      |
> | In-domain                                            | 87.74  | 73.23     | 44.97       | 58.20  | 70.95  | 75.70  |            |
>
> Table 6: The performance of **Prefix-tuning** in few-shot (5%) scenarios across 6 MC datasets.
> _____
> | Trained on $\downarrow$ - Evaluated on $\rightarrow$ | MCTest | ARC(easy) | ARC(chall.) | OBQA   | QASC   | RACE   |
> |:----------------------------------------------------:|:------:|:---------:|:-----------:|:------:|:------:|:------:|
> | MCTest                                               | n/a    | 67.79     | 42.34       | 42.80  | 50.86  | 73.06  |
> | ARC(easy)                                            | 84.76  | n/a       | 40.36       | 46.40  | 48.70  | 72.26  |
> | ARC(chall.)                                          | 82.74  | 59.26     | n/a         | 49.20  | 45.68  | 73.15  |
> | OBQA                                                 | 76.67  | 51.56     | 40.78       | n/a    | 44.17  | 72.98  |
> | QASC                                                 | 81.90  | 73.91     | 38.91       | 49.20  | n/a    | 71.66  |
> | RACE                                                 | 85.60  | 70.54     | 42.62       | 44.20  | 52.59  | n/a    |
>
> In few-shot scenarios, MPrompt's advantage over Prefix-tuning is more pronounced. Specifically, when dealing with 1% and 5% data samples, MPrompt showed improvements of 2.15% and 2.72%, respectively, compared to Prefix-tuning.
> Training with a few samples allows for the adjustment of multi-level prompts, especially domain-specific prompts, to better adapt to the distribution of new datasets and consequently enhance performance.
>
> ## For your concern about the efficiency of MPrompt (as well as QuestionB).
>
>
> Table 7: Comparison of Training Resource Costs for Different Methods.
> _____
> | base          | Trainable parameters | Training time cost per epoch | Test time cost per epoch |
> |:-------------:|:--------------------:|:----------------------------:|:------------------------:|
> | fine-tuning   | 222M                 | 117s                         | 26s                      |
> | prompt-tuning | 0.03M                | 82s                          | 33s                      |
> | prefix-tuning | 29M                  | 88s                          | 34s                      |
> | XPrompt       | 0.03M                | 306s (82s+137s+87s)          | 33s                      |
> | MPrompt       | 39M                  | 92s                          | 36s                      |
>
> We have supplemented our study with a comparison of the training costs between MPrompt and the baseline, as illustrated in Table 7.
> Though XPrompt (an improvement over Prompt-tuning) maintains the same number of trainable parameters as Prompt-tuning, it increases the training time manifold, owing to the three-phase training process inherent in XPrompt.
> While our method (MPrompt) increases both the trainable parameters and training time relative to Prefix-tuning, it yields a 2.25% improvement on 12 datasets of machine reading comprehension. We believe that an additional cost of this magnitude is acceptable.
>
> ## For your concerns about instance-level Prompt and task-specific Prompt.
> Thank you for your valuable comments. Our algorithm utilizes three levels of information which are named task-specific, domain-specific, and context-specific. Each level of information carries different roles in our algorithm and contributes to the overall performance differently.
> Our Figure 2 and Table 8 below provides some supplementary information to illustrate the impact of having different level of prompts in the system.
>
> Table 8: Comparison of the effects of different level prompts (A supplementary to the ablation study in the paper)
> _____
> |       | SQuAD2 | DROP   | OBQA   | BoolQ-NP |
> |:-----:|:------:|:------:|:------:|:--------:|
> | t     | 71.45  | 35.72  | 55.00  | 76.31    |
> | d     | 71.22  | 37.85  | 53.30  | 76.62    |
> | c     | 71.67  | 37.23  | 55.00  | 76.12    |
> | d+c   | 72.06  | 40.79  | 56.60  | 76.23    |
> | t+d+c | 72.61  | 41.64  | 58.20  | 77.25    |
>
> As gleaned from Table 8, we notice that the standalone use of task-specific, domain-specific, and context-specific prompts does not lead to sustained growth. This is partially attributable to the idiosyncrasies of the dataset.
> Task-specific, domain-specific, and context-specific prompts are essentially three fine-grained prompts that focus on different aspects of the data. We are motivated by the fact that there is no guarantee that a particular fine-grained prompt will be superior, and therefore provide information at different levels of granularity to the PLMs.
> Therefore, how to construct multi-level prompt and how to constrain the learning of prompt is what we want to focus on in our work.
>
>
> Thank you once again, for your constructive criticism and valuable insights. Your inputs have greatly helped us refine our paper, and we've done our best to incorporate your suggestions in this revised version. We eagerly await your further recommendations.
>
> Kind regards.

---

### Official Review · Reviewer_2zHj · 2023-08-05

**Typos Grammar Style And Presentation Improvements:** Not found.
**Soundness:** 4

**Excitement:**

4: Strong: This paper deepens the understanding of some phenomenon or lowers the barriers to an existing research direction.

**Missing References:**

1. IDPG: An instance-dependent prompt generation method
2. Instance-wise prompt tuning for pretrained language models

**Paper Topic And Main Contributions:**

The paper discusses about using multi-level prompt tuning to improve the fine-tuning results of PLMs on various MRC tasks. In detail, a soft prompt generator is used to leverage the task, domain and context information into prompt representations. In experimental results, the proposed MPrompt approach surpasses other fine-tuning and prompt tuning baselines. Comprehensive analysis about the detailed settings of prompt tuning are also provided.

**Questions For The Authors:**

1. Are there any comparisons about training resource cost between MPrompt and these baselines?
2. If a new MRC dataset comes, how to adapt MPrompt on it? (in zero-shot, few-shot or fully supervised scenarios)

**Reasons To Accept:**

1. The motivation of using multi-level information to generate prompt representation is intuitive.
2. The experimental result is solid, with necessary fine-tuning and prompt tuning baselines provided and compared. 12 different MRC tasks in various paradigms are considered.
3. The analysis about prompt tuning hyper-params for MPrompt is sufficient.

**Reasons To Reject:**

1. More case studies are recommended to be provided, better along with the visualization result.

**Reproducibility:**

5: Could easily reproduce the results.

**Reviewer Confidence:**

4: Quite sure. I tried to check the important points carefully. It's unlikely, though conceivable, that I missed something that should affect my ratings.

---

> ### Author Rebuttal · Authors · 2023-08-28
>
> Dear Reviewer,
>
> We sincerely thank you for your thorough review and helpful remarks. They have been instrumental in identifying areas where we can enhance the clarity and depth of our paper. We have now revised the paper accordingly and below, we provide detailed responses to your concerns.
>
> ### For your Question1
> Firstly, for the training resource cost, we tested the training cost of each method on the UnifiedQA-base model with the BoolQ dataset as an example, as shown in Table 1.
>
> Table 1: Comparison of Training Resource Costs for Different Methods.
> _____
> | base          | Trainable parameters | Training time cost per epoch | Test time cost per epoch |
> |:-------------:|:--------------------:|:----------------------------:|:------------------------:|
> | fine-tuning   | 222M                 | 117s                         | 26s                      |
> | prompt-tuning | 0.03M                | 82s                          | 33s                      |
> | prefix-tuning | 29M                  | 88s                          | 34s                      |
> | XPrompt       | 0.03M                | 306s (82s+137s+87s)          | 33s                      |
> | MPrompt       | 39M                  | 92s                          | 36s                      |
>
>
> From Table 1, it can be observed that even though XPrompt (an improvement on Prompt-tuning) maintains a consistent amount of trainable parameters, it is accompanied by a significant increase in training time.
> In contrast, our method (MPrompt) achieved an absolute improvement of 2.25% across 12 QA datasets in various paradigms, despite introducing more trainable parameters compared to Prefix-tuning. However, in terms of the order of magnitude of the parameters and training time added, we believe this trade-off to be entirely acceptable.
>
> ### For your Question2
> In response to the second issue, we conducted zero-shot and few-shot experiments across 6 distinct multiple-choice datasets. For the domain labels, we assign the samples in the new datasets to the domain where the nearest cluster center is located.
>
> **For zero-shot scenarios**, we trained optimal MPrompts on each dataset separately and directly inferred on the other datasets.  The results are shown in Tables 2 and 3.
>
> Table 2: The zero-shot performance of **MPrompt** across 6 MC datasets. 'In-domain' refers to the best results achieved through training and testing on the same dataset.
> ____
> | Trained on $\downarrow$ - Evaluated on $\rightarrow$ | MCTest | ARC(easy) | ARC(chall.) | OBQA   | QASC   | RACE   |
> |:----------------------------------------------------:|:------:|:---------:|:-----------:|:------:|:------:|:------:|
> | MCTest                                               | n/a    | 66.67     | 42.49       | 32.80  | 48.06  | 73.40  |
> | ARC(easy)                                            | 77.26  | n/a       | 41.13       | 41.00  | 44.38  | 65.74  |
> | ARC(chall.)                                          | 66.31  | 57.37     | n/a         | 50.00  | 34.34  | 60.79  |
> | OBQA                                                 | 40.12  | 47.47     | 41.13       | n/a    | 20.30  | 40.53  |
> | QASC                                                 | 70.60  | 69.61     | 35.32       | 27.40  | n/a    | 60.52  |
> | RACE                                                 | 85.71  | 67.63     | 43.09       | 34.40  | 50.54  | n/a    |
> | In-domain                                            | 87.74  | 73.23     | 44.97       | 58.20  | 70.95  | 75.70  |
>
> Table 3: The zero-shot performance of **Prefix-tuning** across 6 MC datasets.
> ____
> | Trained on $\downarrow$ - Evaluated on $\rightarrow$ | MCTest | ARC(easy) | ARC(chall.) | OBQA   | QASC   | RACE   |
> |:----------------------------------------------------:|:------:|:---------:|:-----------:|:------:|:------:|:------:|
> | MCTest                                               | n/a    | 61.85     | 39.75       | 32.20  | 48.81  | 73.05  |
> | ARC(easy)                                            | 75.00  | n/a       | 40.02       | 32.60  | 44.95  | 62.28  |
> | ARC(chall.)                                          | 72.14  | 53.62     | n/a         | 48.40  | 31.53  | 59.19  |
> | OBQA                                                 | 40.45  | 49.58     | 40.10       | n/a    | 22.79  | 43.31  |
> | QASC                                                 | 69.17  | 70.50     | 36.95       | 25.60  | n/a    | 66.57  |
> | RACE                                                 | 83.60  | 67.89     | 40.80       | 36.40  | 49.46  | n/a    |
>
> As can be observed from Table 2, MPrompt retains good generalization performance even in zero-shot scenarios, which benefits from semantic prompts at different levels. For example, the model trained on MCTest achieves results comparable to 'In-domain' training on ARC(chall.) and RACE. In comparison with Prefix-tuning (as detailed in Table 3), MPrompt outperforms Prefix-tuning by 0.59% in the zero-shot scenario.
>
> **For few-shot scenarios**, we further fine-tuned the model by randomly drawing 1%, and 5% of the training data from new datasets. The results can be found in Tables 4, 5, 6, and 7, respectively.
>
> Table 4: The performance of **MPrompt** in few-shot (1%) scenarios across 6 MC datasets. '$\uparrow$' denotes the average improvement compared to zero-shot scenarios.
> _____
> | Trained on $\downarrow$ - Evaluated on $\rightarrow$ | MCTest | ARC(easy) | ARC(chall.) | OBQA   | QASC   | RACE   | $\uparrow$ |
> |:----------------------------------------------------:|:------:|:---------:|:-----------:|:------:|:------:|:------:|:----------:|
> | MCTest                                               | n/a    | 66.50     | 42.66       | 41.00  | 48.81  | 73.75  | 1.86%      |
> | ARC(easy)                                            | 80.00  | n/a       | 41.55       | 48.40  | 48.70  | 72.14  | 4.26%      |
> | ARC(chall.)                                          | 80.60  | 57.20     | n/a         | 51.60  | 44.38  | 72.01  | 7.39%      |
> | OBQA                                                 | 42.62  | 49.96     | 41.55       | n/a    | 42.12  | 72.77  | 11.89%     |
> | QASC                                                 | 76.90  | 70.71     | 37.54       | 38.20  | n/a    | 69.43  | 5.87%      |
> | RACE                                                 | 85.60  | 67.63     | 43.17       | 38.80  | 49.89  | n/a    | 0.74%      |
> | In-domain                                            | 87.74  | 73.23     | 44.97       | 58.20  | 70.95  | 75.70  |            |
>
> Table 5: The performance of **Prefix-tuning** in few-shot (1%) scenarios across 6 MC datasets.
> ___
> | Trained on $\downarrow$ - Evaluated on $\rightarrow$ | MCTest | ARC(easy) | ARC(chall.) | OBQA   | QASC   | RACE   |
> |:----------------------------------------------------:|:------:|:---------:|:-----------:|:------:|:------:|:------:|
> | MCTest                                               | n/a    | 63.51     | 42.75       | 37.20  | 48.86  | 73.40  |
> | ARC(easy)                                            | 78.24  | n/a       | 40.02       | 34.20  | 44.84  | 70.98  |
> | ARC(chall.)                                          | 77.26  | 54.00     | n/a         | 48.60  | 43.20  | 71.05  |
> | OBQA                                                 | 41.31  | 50.08     | 40.53       | n/a    | 32.07  | 72.28  |
> | QASC                                                 | 74.48  | 70.12     | 37.71       | 36.60  | n/a    | 67.17  |
> | RACE                                                 | 85.60  | 66.85     | 41.80       | 38.80  | 48.08  | n/a    |
>
>
> Table 6: The performance of **MPrompt** in few-shot (5%) scenarios across 6 MC datasets.
> _____
> | Trained on $\downarrow$ - Evaluated on $\rightarrow$ | MCTest | ARC(easy) | ARC(chall.) | OBQA   | QASC   | RACE   | $\uparrow$ |
> |:----------------------------------------------------:|:------:|:---------:|:-----------:|:------:|:------:|:------:|:----------:|
> | MCTest                                               | n/a    | 69.36     | 42.66       | 51.60  | 57.99  | 73.54  | 6.35%      |
> | ARC(easy)                                            | 84.40  | n/a       | 41.04       | 50.60  | 59.07  | 73.12  | 7.74%      |
> | ARC(chall.)                                          | 83.10  | 65.15     | n/a         | 53.20  | 57.45  | 72.91  | 12.60%     |
> | OBQA                                                 | 84.05  | 51.47     | 42.06       | n/a    | 56.05  | 73.40  | 23.50%     |
> | QASC                                                 | 80.12  | 73.19     | 37.71       | 49.80  | n/a    | 71.66  | 9.81%      |
> | RACE                                                 | 85.00  | 71.68     | 43.52       | 45.80  | 57.67  | n/a    | 4.46%      |
> | In-domain                                            | 87.74  | 73.23     | 44.97       | 58.20  | 70.95  | 75.70  |            |
>
> Table 7: The performance of **Prefix-tuning** in few-shot (5%) scenarios across 6 MC datasets.
> _____
> | Trained on $\downarrow$ - Evaluated on $\rightarrow$ | MCTest | ARC(easy) | ARC(chall.) | OBQA   | QASC   | RACE   |
> |:----------------------------------------------------:|:------:|:---------:|:-----------:|:------:|:------:|:------:|
> | MCTest                                               | n/a    | 67.79     | 42.34       | 42.80  | 50.86  | 73.06  |
> | ARC(easy)                                            | 84.76  | n/a       | 40.36       | 46.40  | 48.70  | 72.26  |
> | ARC(chall.)                                          | 82.74  | 59.26     | n/a         | 49.20  | 45.68  | 73.15  |
> | OBQA                                                 | 76.67  | 51.56     | 40.78       | n/a    | 44.17  | 72.98  |
> | QASC                                                 | 81.90  | 73.91     | 38.91       | 49.20  | n/a    | 71.66  |
> | RACE                                                 | 85.60  | 70.54     | 42.62       | 44.20  | 52.59  | n/a    |
>
>
> As can be observed from Tables 4 and 6, a small amount of data from the new dataset brings substantial benefits. For instance, the MPrompt trained on OBQA achieved an average improvement of 11.89% after receiving 1% of the training set.
> Additionally, compared to the zero-shot scenario, there are improvements of 5.34% and 10.74% for 1% few-shot and 5% few-shot respectively. Most notably, in the scenario of 1% few-shot scenarios, we are already able to find results that are very close to the 'in-domain' training results. Furthermore, MPrompt outperformed Prefix-tuning by 2.15% and 2.72% in the few-shot scenarios with 1% and 5% data samples respectively. This further demonstrates the generalization capabilities of MPrompt.
>
> ### For other concerns about more cases and missing reference papers
> We will add more cases and visualization results in our paper and also include the missing reference papers.
>
> Here is a case:
>
> question: According to cell classification, prokaryotic cells are separated from eukaryotic cells. Which feature is often used to distinguish prokaryotic cells from eukaryotic cells? \n (A) life processes (B) size differences (C) plasma membranes (D) energy molecules
>
> Context: The number and diversity of membranes in eukaryotic cells is a major feature that distinguishes them from prokaryotic cells. A plasma membrane surrounds both eukaryotic and prokaryotic cells. Prokaryotic cells are quite different from eukaryotes. Cell division in eukaryotes differs from that in prokaryotes. Describe two ways in which prokaryotic cells are different from eukaryotic cells. Eukaryotic cells a. differ from prokaryotic cells in that eukaryotes have a backbone. ____ DNA molecules that are used to carry foreign DNA into both prokaryotic and eukaryotic cells. organization of eukaryotic and prokaryotic cells and basic cell processes. Prokaryotic and eukaryotic cells have a cell or plasma membrane. How are prokaryotic cells different from eukaryotic cells?
>
> Prefix-tuning answer: plasma membranes (False answer)
>
> MPrompt answer: size differences (True answer)
>
> In this case, the answer does not explicitly exist in the context, which requires a greater understanding of the context from the model. However, our method, by enhancing the understanding of the context through multi-level prompts, generates the correct answer.
>
>
> In conclusion, we sincerely appreciate your thoughtful comments and advice. We believe that through addressing your concerns, our paper has been significantly improved. We look forward to any further suggestions you may have that can enhance our work.
>
> Best regards.

---

### Official Review · Reviewer_Jx92 · 2023-08-07

**Soundness:** 4

**Excitement:**

3: Ambivalent: It has merits (e.g., it reports state-of-the-art results, the idea is nice), but there are key weaknesses (e.g., it describes incremental work), and it can significantly benefit from another round of revision. However, I won't object to accepting it if my co-reviewers champion it.

**Missing References:**

- Please at least discuss recent parameter efficient tuning techniques:
[0]: https://github.com/huggingface/peft
[1]: https://arxiv.org/abs/2106.09685
[2]: https://arxiv.org/abs/2303.10512
[3]: https://arxiv.org/abs/2210.07558

**Paper Topic And Main Contributions:**

The paper has investigated the incorporation of 2 more prompt vectors designed for the context and domain of the input along with the task prompts for the task of reading comprehension. The technique introduces independence constraints for the domain-specific soft prompts. They conducted experiments on multiple QA datasets to show that the incorporation of  domain and context specific soft prompts is boosting the performance of naive prompt tuning.

**Reasons To Accept:**

- This is a well written paper reporting strong results over multiple QA datasets using prompt-based learning specific for reading comprehension. The technique outperforms previous state-of-the-art methods specific to soft prompt tuning.

**Reasons To Reject:**

- Although the paper successfully discusses and compares against prompt-based learning techniques, it completely ignores other parameter efficient tuning techniques such as LoRA, DyLoRA, and AdaLora. LoRA is a very simple technique that has been shown to outperform prefix-tuning, prompt-tuning and in some cases outperform all fine-tuning. It is not clear how the context-sensitive or domain sensitive vectors could be extended to these recent techniques such as LoRA. The LoRA is simple enough to report its result on your QA datasets.

**Reproducibility:**

4: Could mostly reproduce the results, but there may be some variation because of sample variance or minor variations in their interpretation of the protocol or method.

**Reviewer Confidence:**

5: Positive that my evaluation is correct. I read the paper very carefully and I am very familiar with related work.

---

> ### Author Rebuttal · Authors · 2023-08-28
>
> Dear Reviewer,
>
> Thank you for the time and effort you spent reviewing our paper. We appreciate your valuable feedback and constructive comments which we believe have helped us enrich this work. In the following, we address each of your concerns in detail.
>
> We appreciate your suggestion to compare our method with other parameter efficient tuning techniques, such as Lora. From the inception of this work, we have been primarily focused on investigating the enhancements brought about by Soft Prompts in a specific domain (Machine Reading Comprehension). While both Lora and Soft Prompts belong to parameter efficient tuning techniques, they have slightly different theoretical foundations. Lora is a reparametrization-based method that adds an additional learnable linear layer, while Soft Prompts aim only at modifying the input text to optimize the model. To address your concern, we added comparative experiments with Lora and AdaLora based on the UnifiedQA-base model across 12 QA datasets, as shown in Table 1:
>
> Table 1: Comparison of MPrompt with Lora and AdaLora across 12 QA datasets. '$\uparrow$' denotes the absolute improvement relative to Lora.
> _____
>
> | base       | SQuAD2 | NewsQA  | NarQA   | DROP   | MCTest | ARC(easy) | ARC(chall.) | OBQA   | QASC   | RACE   | BoolQ  | BoolQ-NP |
> |:----------:|:------:|:-------:|:-------:|:------:|:------:|:---------:|:-----------:|:------:|:------:|:------:|:------:|:--------:|
> |            | F1     | ROUGE-L | ROUGE-L | F1     | ACC    | ACC       | ACC         | ACC    | ACC    | ACC    | ACC    | ACC      |
> | Lora       | 75.10  | 47.15   | 45.27   | 19.13  | 86.43  | 68.06     | 43.77       | 36.40  | 47.30  | 73.54  | 82.02  | 65.46    |
> | AdaLora    | 75.11  | 47.14   | 45.26   | 19.14  | 86.42   | 68.06     | 43.77       | 36.40  | 47.30  | 73.54  | 82.02  | 65.53    |
> | MPrompt    | 72.61  | 59.99   | 46.30   | 41.64  | 87.74  | 73.23     | 44.97       | 58.20  | 70.95  | 75.70  | 82.97  | 77.25    |
> | $\uparrow$ | -2.49% | 12.84%  | 1.03%   | 22.51% | 1.31%  | 5.18%     | 1.19%       | 21.80% | 23.65% | 2.16%  | 0.95%  | 11.80%   |
>
>
> From Table 1, it can be observed that Lora does not outperform on most datasets requiring a high level of context understanding. Compared to Lora, MPrompt achieved an 8.49% improvement across 12 datasets. We posit that this might be due to two factors. First, Lora is more suitable for Natural Language Understanding (NLU) tasks as compared to Natural Language Generation (NLG). Second, Lora does not seem to effectively enhance the model's ability to comprehend the context of input. The findings presented in Table 1 further affirm the effectiveness of our proposed MPrompt method, demonstrating its capacity to enhance the model's contextual understanding while only necessitating minor parameter adjustments.
>
> However, we appreciate your insightful suggestion regarding the integration of context-specific and domain-specific information into Lora. We believe this could potentially address the issues we have observed with Lora's performance in Table 1. However, implementing it is not straightforward due to the theoretical differences between Lora and Soft Prompts. We intend to investigate this issue in detail in our future work. Nevertheless, in this paper, our main objective was to explore ways to enhance the performance of Soft Prompts in machine reading comprehension tasks, and our proposed method (MPrompt) showed significant improvement compared to the state-of-the-art methods across 12 QA datasets.
>
>
> In addition, we compared Lora and MPrompt from the perspective of training costs, as detailed in Table 2. Although MPrompt incurs more significant training costs than Lora, considering the performance improvement, we believe this is acceptable.
>
> Table 2: Comparison of Training Resource Costs for Different Methods.
> ___
> | base          | Trainable parameters | Training time cost per epoch | Test time cost per epoch |
> |:-------------:|:--------------------:|:----------------------------:|:------------------------:|
> | fine-tuning   | 222M                 | 117s                         | 26s                      |
> | prompt-tuning | 0.03M                | 82s                          | 33s                      |
> | prefix-tuning | 29M                  | 88s                          | 34s                      |
> | XPrompt       | 0.03M                | 306s (82s+137s+87s)          | 33s                      |
> | Lora          | 0.88M                | 84s                          | 31s                      |
> | AdaLora       | 4.86M                | 131s                         | 38s                      |
> | MPrompt       | 39M                  | 92s                          | 36s                      |
>
>
> To wrap up, we wish to express our deep gratitude, for your detailed review and insightful feedback. Your guidance is instrumental in facilitating our further research. We hope that we have addressed all of your concerns effectively and look forward to any additional comments you might have.
>
> With warm regards.

---

### Meta-Review · Area_Chair_ngmj · 2023-09-18

**Recommendation:** 4

**Metareview:**

The paper introduces Multi-level Prompt Tuning (MPrompt), a method for enhancing the fine-tuning of pre-trained language models (PLMs) for reading comprehension tasks. MPrompt leverages dynamically generated prompts, including domain-specific and context-specific prompts, to improve PLM performance. It demonstrates superior results compared to baseline methods in various MRC datasets and investigates zero-shot and few-shot adaptation scenarios.

Reviewers raised concerns about MPrompt's generalization ability, efficiency, and novelty. They questioned how MPrompt handles new domains, its efficiency compared to baselines, and whether it offers a genuinely novel contribution.  The authors conducted new experiments to showcase MPrompt's generalization ability in zero-shot and few-shot scenarios, compared its efficiency with baselines, and provided evidence supporting the multi-level prompt approach as a valuable contribution. They also acknowledged and addressed the need for further case studies and visualization in the paper, indicating a willingness to improve their work based on reviewer feedback.

**Note:** The paper appears to be closely related to the works https://arxiv.org/pdf/2304.08467.pdf and https://arxiv.org/pdf/2212.10315.pdf, which focus on compressing prompts into soft-prompts. While the papers share similarities in utilizing soft prompts, it would be beneficial for the authors to reference these works and emphasize the distinctions in their approach, particularly in the context of reading comprehension tasks and the utilization of multi-level prompts.

---

### Decision · Program_Chairs · 2023-10-07

**Decision:**

Accept-Findings

**Comment:**

The paper introduces Multi-level Prompt Tuning (MPrompt), a method for enhancing the fine-tuning of pre-trained language models (PLMs) for reading comprehension tasks. MPrompt leverages dynamically generated prompts, including domain-specific and context-specific prompts, to improve PLM performance. It demonstrates superior results compared to baseline methods in various MRC datasets and investigates zero-shot and few-shot adaptation scenarios.

Reviewers raised concerns about MPrompt's generalization ability, efficiency, and novelty. They questioned how MPrompt handles new domains, its efficiency compared to baselines, and whether it offers a genuinely novel contribution.  The authors conducted new experiments to showcase MPrompt's generalization ability in zero-shot and few-shot scenarios, compared its efficiency with baselines, and provided evidence supporting the multi-level prompt approach as a valuable contribution. They also acknowledged and addressed the need for further case studies and visualization in the paper, indicating a willingness to improve their work based on reviewer feedback.

**Note:** The paper appears to be closely related to the works https://arxiv.org/pdf/2304.08467.pdf and https://arxiv.org/pdf/2212.10315.pdf, which focus on compressing prompts into soft-prompts. While the papers share similarities in utilizing soft prompts, it would be beneficial for the authors to reference these works and emphasize the distinctions in their approach, particularly in the context of reading comprehension tasks and the utilization of multi-level prompts.